# Familial Aggregation of Endemic Congenital Hypothyroidism Syndrome in Congo (DR): Historical Data

**DOI:** 10.3390/nu12103021

**Published:** 2020-10-02

**Authors:** Christian X. Weichenberger, Maria Teresa Rivera, Jean Vanderpas

**Affiliations:** 1X Research, Langmoosweg 21, 5023 Salzburg, Austria; research@the-x.net; 2Ecole de Santé Publique, Campus Erasme, Université Libre de Bruxelles, 808 route de Lennik, 1070 Bruxelles, Belgium; mariateresa.rivera@skynet.be

**Keywords:** iodine deficiency, familial aggregation, endemic congenital hypothyroidism, endemic goitre, thyroid metabolism

## Abstract

Familial aggregation of endemic congenital hypothyroidism (CH) in an iodine-deficient population from northern Congo (Democratic Republic (DR)) was analysed on data collected four decades ago (1979–1980). During a systematic survey of 62 families, 46 endemic CH subjects (44 myxedematous and 2 neurological) were identified based on clinical evidence within a village cohort of 468 subjects. A distribution analysis showed that two families presented significant excess of cases versus a random background distribution. Both families were characterised by two healthy parents having all of their five offspring affected by some form of endemic CH. Goitre prevalence in endemic CH was lower than that in the general population, while goitre prevalence in the unaffected part of the cohort (parents and siblings) was similar to that of the general population. Some unidentified genetic/epigenetic factor(s) could contribute to the evolution of some iodine-deficient hypothyroid neonates through irreversible and progressive loss of thyroid functional capacity during early childhood (<5 years old). Besides severe iodine deficiency, environmental exposure to thiocyanate overload and selenium deficiency, factors not randomly distributed within families and population, intervened in the full expression of endemic CH. Further exploration in the field will remain open, as iodine deficiency in Congo (DR) was eliminated in the 1990s.

## 1. Introduction

Endemic congenital hypothyroidism (endemic CH, formerly “endemic cretinism”) affected the most severely iodine-deficient populations as long as iodine supplement was not introduced. In Europe, the syndrome was described until the 1930s (Alps, Pyrenees, Balkans, and Spain) [1,2,3] with last traces reported in Sicily at the end of the 20th century [4]. In other endemic areas of Asia, Central Africa, and Southern America, it was present at least until the end of the 20th century [5,6,7].

Since the incipient description of Himalayan endemic goitre [8], two phenotypes have been associated with severe iodine deficiency: (a) endemic neurological and (b) endemic myxedematous CH. Their clinical expression includes intellectual deficiency in both cases and a picture similar to untreated sporadic congenital hypothyroidism (sporadic CH) in the case of myxedematous phenotype (stunted growth with disproportionate nanism and persistent hypothyroidism) and neurological impairment (spasticity, diplopia, abnormal movements, deaf-mutism, and moderately decreased cranial volume) with no systematic evidence of frank hypothyroidism in the case of neurological phenotype. Pictures of both phenotypes of endemic CH are available, for example, in [5]. The frequency distribution of both phenotypes in the population varies from one endemic region to another. In Central Africa (mainly described in Congo (DR) - Sud-Ubangi province, Uele province and eastern Kivu), the myxedematous type predominates largely [6]. In other countries (Indian Himalaya and New Guinea), the neurological phenotype was largely predominant [5,6,7,8,9]. This classification into two phenotypes is still hypothetic and debated: it has been proposed that most cases have an overlapping syndrome, which associates neurological deficits to an equal degree in both phenotypes and, in addition, myxedematous features resulting from post-natal hypothyroidism in cases defining the myxedematous phenotype [10,11]. According to the work by Stanbury et al. [5], “a striking feature of this form of cretinism (myxedematous) is that the patients are not deaf or mute. Spasticity does not occur except that a Babinski sign is found in about 1/4 of the patients. Reflexes are present but much delayed in relaxation time and squint is unusual”. These clinical criteria certainly apply to most of endemic CH with myxedematous phenotype in Congo (DR), even if clinical assessment conducted by more expert neurologists could likely have revealed more subtle features.

Both phenotypes (neurological and myxedematous) are associated with severe iodine deficiency, with a time-dependent effect [11]:The neurological phenotype is associated with maternal hypothyroidism during the first few months of gestation, when the foetal thyroid gland is not yet functional [11] (autonomous thyroid function in humans appears around the 12th week of gestation [12]).The myxedematous phenotype is associated with hypothyroidism around birth, persisting during the first years of life in the absence of iodine deficiency correction. Infantile and juvenile hypothyroidism in severe iodine deficient areas were quite common. For unknown reasons, some hypothyroid children lose progressively their thyroid responsiveness to iodine, and evolve through irreversible hypothyroidism, even after correction of iodine deficiency [13,14].

Clinically, there are also evident differences in natural sporadic and endemic CH: (a) solely endemic CH is prevented by iodine administered during pregnancy; (b) solely endemic CH hypothyroid children respond to iodine supplements at least during the first four years of life; (c) with the exception of some specific syndromes like Pendred syndrome [15] or thyroid hormone transport defect in X-linked monocarboxylate transporter 8 (MCT8) deficiency [16], untreated sporadic CH is not associated with the particular syndrome of endemic neurological CH.

The causal association of severe iodine deficiency has been well documented with both types of endemic CH: in a not randomised controlled study in New Guinea, iodine supplementation during early pregnancy or before pregnancy decreased markedly the frequency of occurrence of neurological congenital hypothyroidism [17]. For endemic myxedematous, iodised oil during the second trimester of pregnancy [18] or at birth (Dr Philippe Courtois, personal communication) has also been shown to prevent neonatal hypothyroidism in the most severe cases of endemic myxedematous CH [19].

A preferential distribution of cases within certain families has been documented for endemic CH in Ecuador [7] and Sicily [4]. There was also evidence for the health professionals and villagers on the field in Central Africa for familial aggregation—with no numerical data analysis supporting this clinical evidence. By inference from the lack of specific metabolic/enzymatic disorder associated with most cases of sporadic CH of the dysgenesis type (aplasia, hypoplasia, or ectopic thyroids) in industrialised countries, endemic CH is considered as not being associated with a specific enzymatic disorder. In the present study, we describe a pedigree-based analysis of endemic CH of 62 families/468 living subjects conducted in the 1980s in Central Africa in one of the historically most severely affected areas of iodine deficiency, utilizing recently developed statistical analysis tools.

## 2. Materials and Methods

### 2.1. Epidemiological Context

The survey was carried out in 1979 and 1980, in a village at a 20-km distance of Gemena, capital of the Sud-Ubangi province in northern Congo (DR). It was concurrent with the mass distribution of 0.5 mL iodised oil (Lipiodol) containing 240 mg of slowly resorbable iodine in poppy seed oil by intramuscular administration by a mobile medical team (1 medical doctor, 1 medical assistant, 2 nurses, and 1 assistant), all belonging to the Programme de prévention du goitre et du crétinisme. The program was intended to supply the population with iodine every five years in expectation of the transition to iodised salt (effective since 1993). The families were investigated case by case and the structure of the sibship and multigeneration pedigree (in a few cases) was recorded when at least one member of the family was available during the survey.

### 2.2. Clinical Classification

Phenotyping of endemic CH was clinically categorized into two forms:Myxedematous phenotype: moderate to severe intellectual deficiency, stunted growth with disproportionate nanism, persistent myxedema with variable severity, and including mixed form (myxedematous phenotype associated with signs of neurological phenotype);Neurological phenotype: profound intellectual deficiency, no evidence of persistent myxedema, a variable combination of clinical evidence of spasticity, diplopia, abnormal movements, deaf-mutism, and moderately decreased cranial volume.

The diagnosis was posed by a medical doctor (JV) accustomed with this clinical pattern through the public health program of endemic goitre and iodine deficiency disorders prevention. No biosampling was performed in this survey. A more detailed description of the clinical and biochemical characteristics of other subjects with endemic CH in this area has been published elsewhere [19]. In case of a deceased subject in a pedigree, clinical status was considered as undefined for most cases, except for a few subjects for whom the villagers reported that the deceased was effectively affected by the syndrome (“gbamogigi” according to the local term for endemic CH in Ngbaka language). Age was estimated approximately (lack of civil identity document). The presence of goitre was clinically assessed according to the WHO classification [20].

Familial aggregation was investigated for three categories: (1) myxedematous, severe, or moderate myxedematous forms, or a mixture of neurological signs and myxedematous signs; (2) neurological, deaf and/or mute, or neurological handicaps clinically evident of endemic neurological CH [1]; and (3) combined (any of the two phenotypes, either myxedematous or neurological phenotype). This divided study participants into cases and controls: individuals expressing the phenotype were affected and belonged to the group of cases, whereas controls lacked the phenotype.

### 2.3. Basic Statistical Approach

In a first basic approach, the expected probability of endemic CH in each offspring according to the family size (from 1 to 9) for an a priori prevalence of endemic CH in the sibships (10%) was determined. The distribution of observed cases of endemic CH versus expected cases under the random distribution hypothesis within each family size was assessed by the χ^2^-test.

### 2.4. Statistical Development

Pedigree structure and phenotypes were used to statistically assess the observed number and degree of relationship of affected individuals. We utilized tests from the FamAgg Bioconductor R package [21] and the gap R package [22]. Briefly, probability of familial clustering test (Prob. Famil. Cluster., gap R package) and binomial probability test (Binom. Prob., FamAgg R package) make use of counting data only by calculating statistics from the number of cases and family size tabulated in contingency tables. The Prob. Famil. Cluster. test operates on the whole cohort, whereas the Binom. Prob. test is employed family-wise. The Kinship Sum test and the genealogical index of familiality (Genealog. Index) test calculate summary statistics involving the kinship coefficients [23] of related affected individuals. The Kinship Sum test is able to identify cases within families. Originally, the Genealog. Index test was presented as a test comprising the whole cohort [24], and has later been adapted to test single families in FamAgg (termed Fam. Genealog. Index test in this work). For an in-depth assessment of these tests, refer to [25]. Each test returns an empirical *p*-value to observe an outcome at least as extreme as the one found in this study. For Kinship Sum, Genealog. Index, and Fam. Genealog. Index tests, the sampling distribution under the null was obtained by randomly sampling 1,000,000 times the number of affected cases from the cohort. We estimated the exact *p*-value for the Prob. Famil. Cluster. test by 1,000,000,000 sampling steps. Disease prevalence was obtained from the number of affected individuals on a phenotype basis; for the Binom. Prob. test, we additionally investigated common scenarios by setting the prevalence to 2%, 5%, and 7%. Multiple hypothesis testing was addressed by Benjamini–Hochberg correction at a false discovery rate of *p_adj_* = 0.05.

### 2.5. Ethical Aspect

The survey was conducted in accordance with the Ethical Principles for Medical Research Involving Human Subjects defined by the World Medical Association’s Declaration of Helsinki (1979–1980). There was no requirement to obtain individual written consent to accept inclusion in such a family survey or a priori approval for the epidemiological survey by an ethic committee. Practically, it was asked at a community level (village administrative authorities and religious leaders) if the families would accept to give information on their pedigree. To protect the dignity of the participants and their descendants, the location of the surveyed villages remained confidential. The ethical review form of the Erasme University Hospital, Brussels (2019), obtained to analyse data collected in 1979–1980 is included as Document 1: Approval of the ethical committee in the Appendix A.

## 3. Results

### 3.1. Epidemiological Description

The survey included information on 62 families. The family structure was monogamous in 45 cases, bigamous in 12 cases, and polygamous (≥3) in 5 cases. Endemic CH was absent in all mothers and fathers. Overall, the families consisted of 468 clinically phenotyped alive subjects with 46 cases (10%) of endemic CH (Table 1). Most of the cases were of the myxedematous phenotype (severe or moderate in 37 cases, unknown grade in 3 cases, and mixed with neurological signs typical of endemic neurological congenital hypothyroidism in 4 cases). Neurological/deaf-mute phenotype was diagnosed in 2 cases. A total of 162 individuals were reported as deceased at the time of the survey, 3 out of them were reported by the villagers to have had signs of endemic CH.

Median age and goitre prevalence according to the phenotype of presence or absence of endemic CH are described in Table 2 (living subjects). The proportion of goitre, all clinical grades, and visible form was lower in endemic CH than in subjects not affected by endemic CH (Fisher’s exact test (2 tailed), *p* = 0.002 and *p* = 0.005, respectively). The statistical significance should be interpreted cautiously, due to the elevated percentage of missing values in the not-affected cohort (180/422).

### 3.2. Basic Statistical Approach

When analysing the distribution of endemic CH within the sibships (Figure 1), it appeared visually that many points were above the expected number of endemic CH within families stratified according to the family size under the assumption of random distribution with an a priori prevalence of 10% of endemic CH.

To determine if the distribution of expected cases differed from that of the observed cases, the χ^2^-test was performed (Table 3), which led to the rejection of the null hypothesis of a random distribution (*p* = 0.006). The outcome of the χ^2^-test statistics was substantially driven by the two five-membered families with five affected siblings (topmost two points in Figure 1).

### 3.3. Statistical Development

The Genealog. Index and Prob. Famil. Cluster. tests indicated familial aggregation for combined and myxedematous phenotypes with *p*-values < 5 × 10^−6^. These tests are designed to detect familial aggregation on the complete cohort rather than identifying single families. A more detailed analysis utilizing the Kinship Sum test revealed two families with significant aggregation of both myxedematous and combined phenotype (designated A and B, respectively; Table 4). Family A exhibited five individuals with myxedematous phenotype, whereas family B consisted of four individuals with myxedematous phenotype and one female with neurological features (Figure 2).

The findings of the Kinship Sum test were corroborated by the Binom. Prob. test reporting the two families with *p_adj_* ≤ 0.01. For myxedematous phenotype, family B was reported with a borderline *p_adj_* value of 0.05 for tests Binom. Prob. and Kinship Sum. As the only family-based test, Fam. Genealog. Index did not report any significant enrichment of phenotypes in both the families, which was a consequence of family-wise *p*-value calculation specific to this method: the background distribution was retrieved only from the family under investigation, for which in the present case the sampling process resulted in a degenerate distribution that prevented the calculation of an interpretable *p*-value. A detailed summary of all familial aggregation tests performed in this work is presented in Table 4. Phenotype endemic neurological CH occurred only twice in the whole cohort and was observed in two distinct families; by definition, as a standalone trait it does not show any signs of familial aggregation. Due to varying degrees of reported disease prevalence for endemic CH in northern Congo (DR) [6], we used the Binom. Prob. test to investigate different prevalence values. For tested scenarios of lower values of prevalence than that observed in the sample population, additional families were reported as significant with as many as nine families for endemic CH prevalence (2%; Table 5).

Figure 2 shows the pedigrees of the two families with statistical evidence of familial aggregation. Both parents were clinically classified as having no signs of endemic CH in both families. Family B consisted of 8 children (one woman and four men, between ages 6 and 30). There was no palpable goitre in three cases, unimodular palpable goitre with neck in extension in one case, and multinodular voluminous goitre in one case. Two of the five affected subjects were of the mixed phenotype, one was of the myxedematous phenotype, one was of the moderate form of myxedematous phenotype, and one was of the neurological phenotype. Family A consisted of 8 children, of whom 5 were living (2 men and 3 women aged between 5 and 25 years). There was no goitre in one case, not-visible palpable goitre with the neck not extended in 2 cases, and visible goitre with the neck not extended in one case. All cases were of the myxedematous phenotype.

Removal of families A and B from the analysis resulted in the loss of any significant familial aggregation of the combined phenotype for tests Kinship Sum and Binom. Prob., whereas the Genealog. Index test still reported signs of familial clustering (*p* = 0.0008), which was likely caused by another family consisting of 4 affected members and 3 healthy subjects. The observed loss of significance in the hypothetical scenario where families A and B were missing provided further confidence in the numeric stability of Kinship Sum and Binom. Prob. tests and the importance of the two detected families in this study.

## 4. Discussion

Endemic CH differs from sporadic CH by its prevalence: at birth in historical severely affected areas such as northern Congo (DR), neonatal hypothyroidism (cord serum thyroid stimulating hormone (TSH) > 20 mU/L) was documented in 11% of neonates [18]; this value has to be compared with the much lower prevalence of neonatal hypothyroidism in industrialised countries (between 1/2500 and 1/3800 neonates being hypothyroid during neonatal screening for congenital hypothyroidism). The difference is also physiological: endemic CH is prevented by the correction of iodine deficiency (at least, when correction covers the pregnancy since its inception). Sporadic CH requires L-thyroxine (L-T4) therapy since early life to prevent the sequelae of congenital hypothyroidism.

While iodine deficiency is clearly defined as a causal factor of endemic CH, the variable occurrence of cases within families of a close geographical proximity reflects familial aggregation of environmental nature and/or genetic nature. The method of approach described here does not permit to distinguish an environmental or genetic nature of association with the familial aggregation [26]. Up to now, it was not excluded that endemic CH developed randomly within the population and within families, as most subjects present at some time of their development severe hypothyroidism. This vision can now be refuted by the present data: there is/are some factor(s), genetic and/or environmental not randomly distributed, which contribute to the aggregation of cases within some families in the whole population.

Due to the fact that endemic myxedematous CH was associated with decreased iodine uptake and with reduced goitre [27] in comparison with the local iodine-deficient population, it was generally considered that this iodine-deficiency associated phenotype is more closely analogous to sporadic CH with dysgenesis than to sporadic CH with dyshormonogenesis. When looking at the evolution of knowledge in sporadic neonatal hypothyroidism, most of the cases with dysgenesis (aplasia, hypoplasia, and ectopic gland) could not be associated with a specific genetic/metabolic defect—despite the tremendous effort to identify some common metabolic pathway defect in thyroid development. The general concept is that, besides exceptions, sporadic CH with dysgenesis is associated with multifactorial multigenic diversity of genetic or epigenetic [28] defects with low incidence in the population [29,30,31]. By contrast, the present results obtained by the familial aggregation analysis suggest that endemic CH may be in part also of genetic origin—in case of severe iodine deficiency, the data highlighted aggregation at an elevated prevalence in at least two families of the cohort. A genetic factor does not exclude other environmental factors in addition to severe iodine deficiency with strong evidence of effect on thyroid function: thiocyanate overload in this region results from consumption of poorly detoxified cassava and selenium deficiency is linked to geographical lack in the soil [32]. In villagers with a very homogeneous exposure to severe iodine deficiency, goitrogen exposure (such as thiocyanate overload), and increased susceptibility to free radical toxicity (selenium deficiency), some genetic factors might contribute to this not-random distribution of endemic CH within families.

In the randomized clinical trial of Karawa (1 mL iodised oil vs. no iodine during the 2nd trimester of pregnancy, period 1974–1981), the most severe forms of endemic myxedematous CH were prevented in the treated group, their infants being protected of iodine deficiency until weaning [19]. A great proportion of endemic hypothyroid neonates and young children (cord serum or infantile serum TSH > 20 mU/L) [18,19,32] did not evolve through persistent myxedematous, and presented a phase of spontaneously reversible hypothyroidism of variable severity, variable duration, and variable timing, even in the absence of artificial iodine supplementation. Nevertheless, there was a subgroup of children who, for unknown reasons, remained hypothyroid and evolved through the full picture of endemic CH.

Genetic monolocus dominant transmission was excluded, as none of the parents were clinically affected by endemic CH. It is well-known that hypothyroidism reduces fertility [33], and the co-occurrence of mild form of endemic CH within mothers during mass campaigns of iodised oil administration was exceptional (JV, personal observation). Balanced distribution of both sexes in cases of endemic CH excluded transmission by sex chromosome. As already mentioned, goitre was less frequent in endemic CH than in unaffected subjects. The elevated prevalence of goitre in this last group was similar to the prevalence described in a large epidemiological survey at the same period [6]. It was further observed that, besides the cases of endemic CH, their siblings in the same families did not present a different risk of goitre when compared with the general population.

When explaining the results with a monogenetic recessive transmission model, each offspring was affected with a probability of 1/4, such that for family A the probability to observe five living affected offspring was only *p* = (1/4)^5^ = 0.001. Similarly, the probability to observe at least four children with a myxedematous phenotype out of five in family B was computed as *p* = 5 × (1/4)^4^ × (3/4) + (1/4)^5^ = 0.02. Based on these low probabilities and following the law of parsimony, a simple genetic transmission of recessive mono-allelic trait is unlikely.

Besides the clinical ascertainment bias already described in young children, familial aggregation analysis is not sufficient to establish a causal relationship between genetics and a clinical syndrome [26]. Molecular genetics would be required for further investigations [34,35,36]. Fortunately, iodine deficiency is controlled since that epoch: after the first programs of iodised oil administration (1974–1995), obligatory iodised salt commercialisation has been progressively implemented since 1993 in northern Congo (DR) and monitoring of endemic goitre [37,38] reveals the resolution of this public health major problem over the whole country. Scientifically, the question of an underlying genetic mechanism will remain open.

### Major Limits of the Investigation

In the present work, the definition of endemic CH was exclusively based on clinical expertise, without thyroid function tests, thyroid scintigraphy, echography, or radiological assessment of osseous development. This clinical classification of endemic CH in this study was identical to the one used during mass distribution of iodised oil to control iodine deficiency disorders in this region [6] and was applied by a large group of medical teams on the field. A more detailed clinical profile of endemic CH obtained in other surveys at the same period is described in reference [19]. Cases and controls were classified exclusively by clinical ascertainment. Below the 5-year-age, classification should be considered as uncertain: children less than 5 years old definitely affected or transiently affected by hypothyroidism should present a similar pattern of clinical signs. This could be a source of bias of small scale in the present description, with 5 of the 44 endemic CH appertaining to this age group. Moreover, it is known that the clinical sensitivity of hypothyroidism is low in infants: it was known, at the early stage of sporadic CH screening program, that the clinical examination alone, in absence of measurement of thyroid function hormones, had a sensitivity of 10% at birth and 70% in one-year-old children even by experienced paediatricians [39].

The survey was limited to a unique village. This might raise questions about the representativeness of the results in the whole iodine-deficient geographical area covering more than 4 million people. It had been well-established that even in the same endemic area, villages were variably affected: for example, endemic goitre and endemic CH prevalence were much lower in the villages along the rivers (locally named “gens d’eau” translation: “waterpeople”) [6]. The present study was not a general population study and therefore exhibited a certain degree of ascertainment bias, since the endemic CH proportion in severely affected villages in [6] extended between 1.7% and 9% and our proportion observed in the village of the familial aggregation was 10%. The village presented in this work was obviously selected for its elevated endemic CH proportion.

No biomolecular analysis was performed to validate the hypothesis. Such an absence of validation will likely remain unsolved as iodine deficiency has been corrected since more than 25 years in this area.

## Figures and Tables

**Figure 1 nutrients-12-03021-f001:**
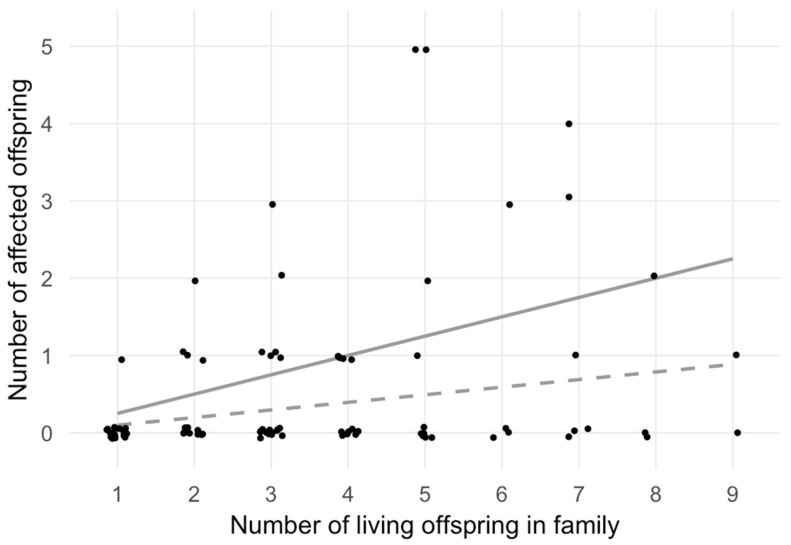
Number of affected offspring versus total number of offspring. This scatter plot shows the number of living offspring (*x*-axis) and the number of affected offspring (*y*-axis) within each nuclear family. Polygamic family structures and families with multiple generations are split into nuclear families with dedicated father and mother and their associated offspring. Data points are plotted slightly off-value in order to increase visibility. The two grey lines illustrate the expected number of affected individuals by family size, based on a random disease prevalence of 10% (dashed line) and on a monogenic Mendelian recessive trait resulting in a 25% chance to become an affected sibling (continuous line). Families A and B (Figure 1) are clearly visible as the two topmost points, since all five family members were affected.

**Figure 2 nutrients-12-03021-f002:**
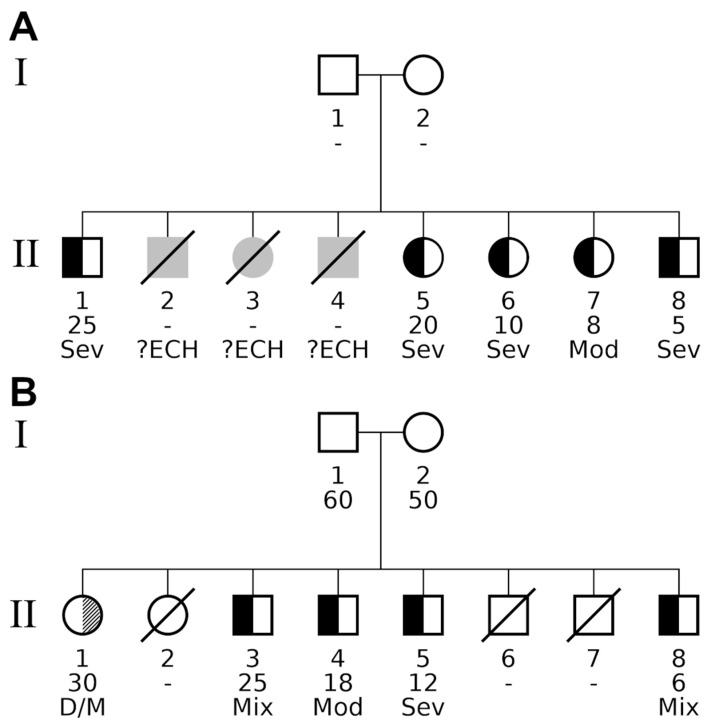
Pedigrees of nuclear families detected by various familial aggregation methods. The pedigrees contained two types of diseases, indicated by split family member symbols. Black fillings in the left half of a subject’s symbol indicate individuals affected by the myxedematous phenotype. A diagonally hatched right-hand segment designates individuals exhibiting the neurological phenotype. Unaffected individuals have a white (i.e., empty) segment encoding the respective phenotype. Additional information is given below a pedigree symbol. 1st line: participant identifier (unique for each generation); 2nd line: age at study participation, unknown age is specified by a hyphen (“-“). Offspring are ordered from left to right by decreasing age; 3rd line: detailed description of the phenotype as follows: Sev (severe) or Mod (moderate) form of persistent myxedema; ?ECH, undefined phenotype of CH; D/M, deaf/mute; Mix, mixed form of myxedematous and neurological phenotype. Both families had healthy parents. (**A**) Significant myxedematous phenotype aggregation within family A. All living offspring showed signs of myxedema, and even diseased individuals II.2, II.3, and II.4 were reported by villagers to have had suffered from some form of endemic CH (indicated by grey symbols). The late subjects were born between individuals II.1 and II.5, and would have had an age between 20 and 25 years at the date of survey. Overall, family A consisted of eight affected offspring, five alive and three deceased, all of them affected by endemic CH. (**B**) Significant neurological and myxedematous aggregation in family B. In this family, individual II.1 exposed the neurological phenotype, so when investigating the combined phenotype for familial aggregation, family B had exactly the same number of affected alive offspring and was reported with the same *p*-values by all three family-based tests. Looking at the myxedematous phenotype only revealed the major difference compared with family A: four affected individuals (II.3, II.4, II.5 and II.8) gave rise to a borderline significant result for tests Binom. Prob. and Kinship Sum (*p_adj_* = 0.05 in both tests).

**Table 1 nutrients-12-03021-t001:** Phenotype classification of the subjects with endemic congenital hypothyroidism (CH) during the survey on familial aggregation in Ubangi, northern Congo (Democratic Republic) (1979–1980).

Normal	Myxedematous (*n* = 44)		Neurological (*n* = 2)		Dead (*n* = 162)		Unknown (*n* = 4)
422	Grade Severe	20	Neurological	1	Without signs ^1^	159	4 ^2^
	Moderate	17	Deaf/mute	1	With signs ^1^	3	
	Unknown grade	3					
	Mixed phenotype	4					

^1^ Endemic CH reported by the villagers. ^2^ These individuals were not counted in any statistics.

**Table 2 nutrients-12-03021-t002:** Characteristics of subjects with and without endemic CH.

		Endemic CH (*n* = 46)	No Endemic CH (*n* = 422)	Test and 2-Tailed *p*-Value ^3^
**Gender**	Number of M/F (ratio M/F)	24/22 (1.09)	185/237 (0.78)	Fisher’s *p* = 0.35
**Age**	Missing values, *n*/total (%)	2/46 (4.3%)	136/422 (32.23%)	Fisher’s *p* = 1.3 × 10^−5^
	Median, years (IQR) (*n* obs ^1^)	15.5 (6.5, 23.0) (44)	16.0 (6.0, 30.0)(286)	Mann–Whitney’s *p* = 0.33
	<5 years, *n*/*n* obs (%)	5/44 (11.36%) ^2^	51/286 (17.83%)	
	5–18 years, *n*/*n* obs (%)	19/44 (43.18%)	99/286 (34.62%)	
	≥18 years, *n*/*n* obs (%)	20/44 (45.46%)	136/286 (47.55%)	Fisher’s *p* = 0.46
**Goitre**	Missing values, *n*/total (%)	4/46 (8.7%)	180/422 (42.65%)	Fisher’s *p* = 1.9 × 10^−6^
	Any form, *n*/*n* obs (%)	22/42 (52.38%)	187/242 (77.27%)	Fisher’s *p* = 0.002
	Visible form, *n*/*n* obs (%)	18/42 (42.86%)	161/242 (66.53%)	Fisher’s *p* = 0.005

^1^*n* obs, number of observations. ^2^ Percentages refer to the respective stratum described by the column. ^3^ Fisher’s *p* by Fisher’s exact test; Mann–Whitney’s *p* by Mann–Whitney test. M/F, Male/Female; IQR, interquartile range.

**Table 3 nutrients-12-03021-t003:** Basic statistical analysis of the distribution of observed number of endemic CH versus expected number under random distribution within each nuclear family according to the sibship size.

Sibship size	1	2	3	4	5	6	7	8	9
Total no. families ^1^	19	15	19	11	10	4	6	3	2
Affected no. families ^1^	1	5	6	4	4	1	3	1	1
Observed no. ^2^	1	5	9	4	13	3	8	2	1
Expected no. ^3^	1.87	2.95	5.60	4.32	4.91	2.36	4.13	2.36	1.77
χ^2^-test statistics	χ^2^ = 21.41, df = 8, *p* = 6.13 × 10^−3^								

^1^ Number of nuclear families with respective sibship size. ^2^ Number of endemic CH cases within affected families according to sibship size. ^3^ Expected number of affected siblings according to sibship size is based on the village’s disease prevalence, 46/468 ≈ 10% (Table 2). df, degree of freedom.

**Table 4 nutrients-12-03021-t004:** Statistical analysis of the distribution of endemic CH familial aggregation within two families and within the whole population.

		Family A		Family B	
Phenotype	Test	*p*	*p_adj_*	*p*	*p_adj_*
Combined (neurological and myxedematous)	Binom. Prob. ^1^	1.62 × 10^−4^	5.04 × 10^−3^	1.62 × 10^−4^	5.04 × 10^−3^
	Fam. Genealog. Index	0.523	1	0.523	1
	Kinship Sum	1.95 × 10^−3^	8.98 × 10^−3^	1.95 × 10^−3^	8.98 × 10^−3^
	Population based distribution of endemic CH within all families				
	Genealog. Index ^2^	*p* < 1.00 × 10^−6^			
	Prob. Famil. Cluster. ^2^	*p* = 5.05 × 10^−21^			
Myxedematous	Binom. Prob.	9.31 × 10^−5^	5.77 × 10^−3^	1.66 × 10^−3^	5.14 × 10^−2^
	Fam. Genealog. Index	0.524	1	0.714	1
	Kinship Sum	1.23 × 10^−3^	1.01 × 10^−2^	1.14 × 10^−2^	5.20 × 10^−2^
	Population based distribution of endemic CH within all families				
	Genealog. Index	5.00 × 10^−6^			
	Prob. Famil. Cluster.	4.32 × 10^−7^			

^1^ Reported values of the Binom. Prob. test refer to 10% prevalence as derived from the sample population (Table 2). ^2^ Tests Genealog. Index and Prob. Famil. Cluster. operate on the whole population rather than individual families, a correction for multiple hypothesis testing is not applicable.

**Table 5 nutrients-12-03021-t005:** Observed and tested disease prevalence scenarios for the binomial probability test.

	Observed	Tested Scenarios		
Disease prevalence	10% ^1^	7%	5%	2%
	Number of observed families with endemic CH aggregation at *p_adj_* < 0.05			
Combined phenotype	2 (families A, B)	3	3	9
Myxedematous phenotype	1 (family A)	2	2	9

^1^ Disease prevalence as derived from the sample population, as reported in Table 2.

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
