# Peer review of "Familial Aggregation of Endemic Congenital Hypothyroidism Syndrome in Congo (DR): Historical Data"

_nutrients, 2020, doi:10.3390/nu12103021_

Round 1

Reviewer 1 Report

This is an interesting, if very unusual piece of research, reanalysing data collected more 40 years ago. The question is "what motivated this research"? The problem this reviewer has had has been the lack of access to examine the details of the original data to confirm the information on which individuals were placed into diagnostic categories.References to the original data are obscure. For example, I cannot find reference 28 and references six, 10, 20, are not readily available. There are errors in what is provided for references 19 and 28 and reference 33 is a thesis written in French and available only from the Libre de Bruxelles. How does one obtain access to documents such as this within the short timeframe allocated for providing a peer review? This is unacceptable. The best information I can find is in the Stanbury et al publication (reference 5) in which he states; "the central difficulty in any consideration of endemic cretinism is its definition and ------ the diagnosis can never be unequivocal" and this is the issue central to this current piece of research. If we accept that neurological cretinism is caused by intrauterine factors (hypothyroxinemia) secondary to maternal iodine efficiency and that myxedematous cretinism is secondary to postnatal hypothyroidism (Reference 11), we need to be reassured that the original population was subject to expert neurological examination and was correctly classified. Again, returning to the Stanbury article he says that "a striking feature of this form of cretinism (myxedematous) is that the patients are not deaf or mute. Spasticity does not occur except that a Babinski sign is found in about 1/4 of the patients. Reflexes are present but much delayed in relaxation time and squint is unusual". Therefore, as there was no convincing evidence of prenatal hypothyroidism in these subjects it was a most likely that these afflicted persons were suffering from postnatal hypothyroidism with atrophic thyroid glands as a primary cause and not secondary to other unlikely putative causes such as thiocyanate intoxication or selenium deficiency as your group has proposed in the past. Further in the

46 endemic CH subjects, 40 forward myxedematous and only 2 were said to be neurological. I'm not aware of anywhere else in the world where endemic iodine deficiency causes such a distribution. Is there anywhere else to your knowledge where this occurs.
My question is do you genuinely believe that the underlying problem is identification see with some superimposed factor causing thyroid atrophy or is it possible that the original diagnoses were incorrect? While arguments have been put forward thiocyanate overload and/ selenium deficiency is there any good evidence for these factors causing thyroid atrophy? Did the myxedematous form of cretinism disappear with the correction of iodine deficiency in these communities? It would be a great contribution to clear up this issue that has plagued this field of research for over 40 years.

Please correct the errors in your references and replace those cited that they are not in peer reviewed journals.

Author Response

REVIEWER 1

This is an interesting, if very unusual piece of research, reanalysing data collected more 40 years ago. The question is "what motivated this research"? The problem this reviewer has had has been the lack of access to examine the details of the original data to confirm the information on which individuals were placed into diagnostic categories.

  1. We have put more emphasis in the “Limitation” paragraph on the fact that our clinical classification was based on the same approach as the one used by many MD’s during epidemiological survey in Northern Congo. See lines 349-353 in the revised manuscript and also our answer [3.] hereunder.

References to the original data are obscure. For example, I cannot find reference 28 and references six, 10, 20, are not readily available. There are errors in what is provided for references 19 and 28 and reference 33 is a thesis written in French and available only from the Libre de Bruxelles. How does one obtain access to documents such as this within the short timeframe allocated for providing a peer review? This is unacceptable.

  1. We are sorry for the confusion that has been created. We have modified the references as follows:

    • Ref. 6: a URL link has been added to get direct access to IDRC monograph to facilitate the access to the reference.
    • Initial 10 DeLong GR 1989 has been removed as the information is already present in the more easily accessible reference current ref. 10 Cao XY … DeLong GR NEJM 1994.
    • Initial 19 current ref. 18 Thilly & al. 1978: mistake in this reference has been corrected (47(2) in place of 36).
    • Initial 28 current ref. 27 Delange 1972 J Clin Endocrinol Metabol in place of J Endocrinol.
    • Initial 20 current ref. 19 Vanderpas 1994, which is not easily accessible: please find attached a PDF of this document. A permission to add this chapter as “Supplementaty material” to the manuscript has been obtained from the Publisher Cognizant Communication Corporation.

The best information I can find is in the Stanbury et al publication (reference 5) in which he states; "the central difficulty in any consideration of endemic cretinism is its definition and ------ the diagnosis can never be unequivocal" and this is the issue central to this current piece of research. If we accept that neurological cretinism is caused by intrauterine factors (hypothyroxinemia) secondary to maternal iodine efficiency and that myxedematous cretinism is secondary to postnatal hypothyroidism (Reference 11), we need to be reassured that the original population was subject to expert neurological examination and was correctly classified. Again, returning to the Stanbury article he says that "a striking feature of this form of cretinism (myxedematous) is that the patients are not deaf or mute. Spasticity does not occur except that a Babinski sign is found in about 1/4 of the patients. Reflexes are present but much delayed in relaxation time and squint is unusual". Therefore, as there was no convincing evidence of prenatal hypothyroidism in these subjects it was a most likely that these afflicted persons were suffering from postnatal hypothyroidism with atrophic thyroid glands as a primary cause and not secondary to other unlikely putative causes such as thiocyanate intoxication or selenium deficiency as your group has proposed in the past.

  1. We have addressed this point in the manuscript in lines 50-55 by quoting from the work of late John B Stanbury:

    According to the work by Stanbury et al. [5]: "a striking feature of this form of cretinism (myxedematous) is that the patients are not deaf or mute. Spasticity does not occur except that a Babinski sign is found in about 1/4 of the patients. Reflexes are present but much delayed in relaxation time and squint is unusual”. These clinical criteria certainly apply to most of endemic CH with myxedematous phenotype in Congo (RD), even if clinical assessment conducted by more expert neurologists could likely have revealed more subtle features.

Further in the 46 endemic CH subjects, 40 forward myxedematous and only 2 were said to be neurological. I'm not aware of anywhere else in the world where endemic iodine deficiency causes such a distribution. Is there anywhere else to your knowledge where this occurs.

  1. To our knowledge, this point (relative proportion of NEURO phenotype & MYX phenotype) has been addressed in New Guinea and Ecuador: neurological phenotype was largely predominant. In Northern China, 1/3 myxedematous phenotype, 2/3 neurological phenotype. Why was there such a predominance of myxedematous phenotype in Eastern & Northern Congo? There is no clear answer. As you say, this is a 40 year old source of confusion and discussion, and the present paper is not able to resolve this point, either. We intend only to add some more information: endemic CH is not randomly distributed, there is familial aggregation, and in accordance to the recent developments of thyroid biochemistry (example: DUOX variation in human population modifying the susceptibility to iodine deficiency), a potential genetic factor could explain – maybe – such a genetic driven susceptibility to iodine deficiency in some children. We do not know whether endemic CH occurs still in other iodine deficient areas, but if yes, it seems pertinent to associate an iodine deficiency control program with a more academic question on genetic approach, covering the full range of nowadays ethical requirements.

My question is do you genuinely believe that the underlying problem is identification see with some superimposed factor causing thyroid atrophy or is it possible that the original diagnoses were incorrect?

  1. We think that our clinical definition was homogeneous between various MDs intervening in the public health program in Congo (RD), even if limited to evident neurological dysfunction as described hereabove by John B Stanbury. This is now included in the revised manuscript (lines 50-55).

While arguments have been put forward thiocyanate overload and/ selenium deficiency is there any good evidence for these factors causing thyroid atrophy?

  1. This is a vast subject, and much work has been published on these environmental factors and experimental model. For the sake of simplicity, we have only mentioned environmental factors (lines 24; 313- 314). According to the Rothman compartment cause model, potential genetic factors do not exclude environmental factors. In the absence of biomolecular data, we have chosen to refrain mentioning gene x environment interactions, as this does not add value to the work presented here.

Did the myxedematous form of cretinism disappear with the correction of iodine deficiency in these communities? It would be a great contribution to clear up this issue that has plagued this field of research for over 40 years.

  1. Yes, clearly, iodised oil to the mother during the second trimester of pregnancy prevents endemic CH – myxedematous phenotype – in Northern Congo (RCT Karawa study): look at Figure 5 of attached current reference 19 (Vanderpas 1994). This has also clarified in the manuscript (lines 314-316).

    Please correct the errors in your references and replace those cited that they are not in peer reviewed journals.

We have fixed this as indicated above (Our answer [2.]).

Reviewer 2 Report

The study focus on familial aggregation of endemic congenital hypothyroidism in iodine deficient Northern Congo using the historical sample from 1979-1980

Table 3: There are 180 missing values out of 422 subjects in control group.(42.65%). I don’t agree with the interpretation that the proportion of goiter was lower in the endemic CH group compared with the control group.

The authors attempted to find any genetic mechanism involved in endemic CH using family aggregation analysis, even though it's well known that molecular genetics is ideal. It cannot be conducted in that area since CH has been eradicated in the region of Congo.

The major limitation to this study is diagnosis of CH only based on clinical exam. The prevalence of CH may be significantly higher than reported in the study. Also the authors used only 1 village, the results cannot be generalized to the whole region of Northern Congo.

Author Response

REVIEWER 2

The study focus on familial aggregation of endemic congenital hypothyroidism in iodine deficient Northern Congo using the historical sample from 1979-1980

  1. Table 3: There are 180 missing values out of 422 subjects in control group.(42.65%). I don’t agree with the interpretation that the proportion of goiter was lower in the endemic CH group compared with the control group.

    We agree with this remark: a phrase has been added (lines 172-173):

    The statistical significance should be interpreted cautiously, due to the elevated percentage of missing values in the not affected cohort (180/422).

The authors attempted to find any genetic mechanism involved in endemic CH using family aggregation analysis, even though it's well known that molecular genetics is ideal. It cannot be conducted in that area since CH has been eradicated in the region of Congo.

The major limitation to this study is diagnosis of CH only based on clinical exam. The prevalence of CH may be significantly higher than reported in the study. Also the authors used only 1 village, the results cannot be generalized to the whole region of Northern Congo.

  1. There is, indeed, great geographic variation of goiter and endemic CH: this information has been added on lines 364-367:

    It is well established that even in the same endemic area, villages were variably affected: for example, endemic goiter and CH prevalence were much lower in the villages along the rivers (locally named “gens d’eau” translation: “waterpeople”) [6].

    Even though we have sampled a whole village, this study is not a general population study and therefore exhibits a certain degree of assortment bias, since the endemic CH proportion in severely affected villages in reference 6 extends between 1.7% and 9% and our proportion observed in the village of the familial aggregation was 10%. The village presented in this work was obviously selected for its elevated endemic CH proportion. This information has been added in the paragraph 4.1., lines 365-369.